# Accurate Retraining-free Pruning for Pretrained Encoder-based Language Models

**Seungcheol Park**[1]**, Hojun Choi**[2]\***& U Kang**[1]†
[1]Seoul National University, Seoul, South Korea
[2]Kim Jaechul Graduate School of AI, KAIST, Seoul, South Korea
{ant6si, ukang}@snu.ac.kr, hchoi@kaist.ac.kr

## Abstract

Given a pretrained encoder-based language model, how can we accurately compress it without retraining? Retraining-free structured pruning algorithms are crucial in pretrained language model compression due to their significantly reduced pruning cost and capability to prune large language models. However, existing retraining-free algorithms encounter severe accuracy degradation, as they fail to handle pruning errors, especially at high compression rates. In this paper, we propose K-prune (Knowledge-preserving pruning), an accurate retraining-free structured pruning algorithm for pretrained encoder-based language models. K-prune focuses on preserving the useful knowledge of the pretrained model to minimize pruning errors through a carefully designed iterative pruning process composed of knowledge measurement, knowledge-preserving mask search, and knowledge-preserving weight-tuning. As a result, K-prune shows significant accuracy improvements up to 58.02%p higher F1 score compared to existing retraining-free pruning algorithms under a high compression rate of 80% on the SQuAD benchmark without any retraining process.

## 1 Introduction

*How can we accurately compress pretrained encoder-based language models without retraining?* Transformer-based PLMs dominate (Devlin et al., 2019; Clark et al., 2020; Liu et al., 2019; Brown et al., 2020; Zhang et al., 2022) the field of Natural Language Processing (NLP) based on their remarkable performance. The superiority of PLMs comes with a massive increase in their size, and the unaffordably scaled models necessitate compression algorithms that effectively reduce the size of PLMs without compromising accuracy.

Retraining-free structured pruning algorithms (Kwon et al., 2022b; Nova et al., 2023) are prominent for compressing pretrained language models (PLMs) since they require dramatically lower computational costs and a smaller amount of data than existing retraining-based algorithms (Hou et al., 2020; Liu et al., 2021; Lin et al., 2020; Wang et al., 2020b; Sajjad et al., 2023; Xia et al., 2022; Lagunas et al., 2021). Retraining-free algorithms achieve remarkable efficiency by replacing an expensive retraining process with a one-shot mask search process followed by a lightweight mask-tuning process. However, when it comes to the high compression rate, retraining-free algorithms exhibit severe accuracy degradation. The accuracy degradation comes from a failure of handling pruning errors which represent the distortion of the model's prediction by the accumulated deformations of the outputs of the pruned intermediate layers.

In this paper, we propose K-prune (Knowledge-preserving pruning), an accurate retraining-free structured pruning algorithm for encoder-based PLMs. We conceptualize pruning error as the loss of useful knowledge to explicitly measure the amount of pruning error. We observe that the main reason of severe accuracy degradation in previous retraining-free pruning algorithms is an unrecoverable knowledge loss from multiple layers. Therefore, we carefully design an iterative pruning process that distributes the knowledge loss across multiple iterations to overcome the accuracy degradation

---

\*Work done while at Seoul National University
†Corresponding author

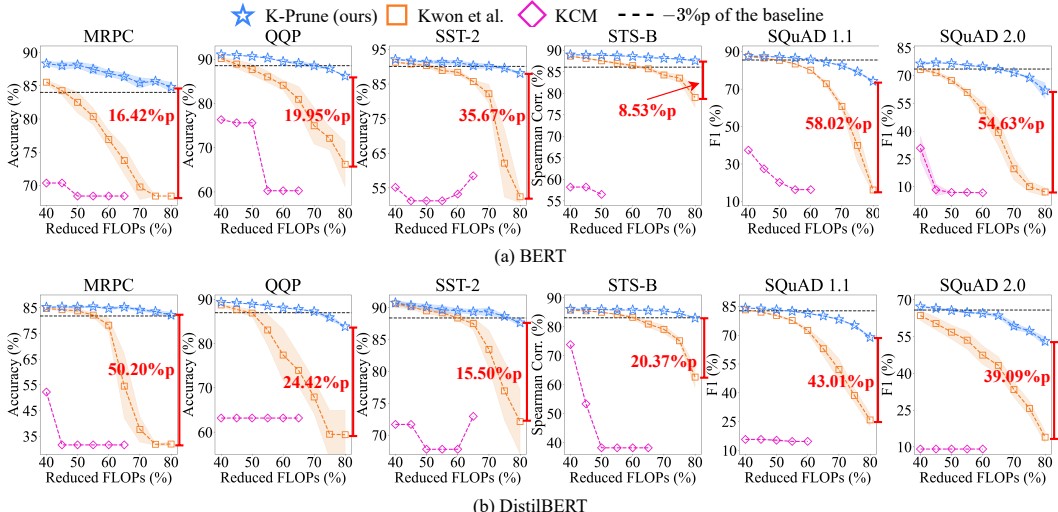

Figure 1: Accuracy vs. reduced FLOPs of retraining-free pruning algorithms using BERT and DistilBERT where the dotted line indicates the accuracy degradation of 3%p. K-prune (blue star) largely outperforms competitors in all settings.

problem. Our iterative pruning process consists of three steps which aim to preserve the model's useful knowledge: (1) knowledge measurement, (2) knowledge-preserving mask search, and (3) knowledge-preserving weight-tuning. Our iterative pruning is different from previous retraining-based iterative pruning approaches (Frankle & Carbin, 2019; Han et al., 2015) since K-prune systemically controls the degree of pruning in each iteration. K-prune efficiently prunes the pretrained language models by an efficient weight-tuning technique which runs within a second requiring only a small sample dataset. As a result, K-prune successfully overcomes the accuracy degradation problem and shows up to 58.02%p[1] higher F1 score compared to the other retraining-free pruning algorithms as depicted in Figure 1. We summarize our main contributions as follows:

- **Algorithm**. We propose K-prune, an accurate retraining-free pruning algorithm for PLMs. K-prune consists of three novel ideas to preserve the useful knowledge of the pretrained models: knowledge measurement, knowledge-preserving mask search, and knowledge-preserving weight-tuning.
- **Accuracy**. We perform extensive experiments on GLUE and SQuAD benchmarks to demonstrate the performance of K-prune. K-prune shows up to 58.02%p higher F1 score than the best results of existing retraining-free algorithms under a high compression rate of 80%.
- **Efficiency**. We demonstrate that K-prune shows the best accuracy-cost trade-off among the state-of-the-art pruning algorithms. K-prune shows comparable or higher accuracy than retraining-based algorithms on GLUE benchmarks with up to 422× lower pruning cost.

Our source code is available at `https://github.com/snudm-starlab/K-prune`

## 2 PRELIMINARIES

### 2.1 ENCODER-BASED PRETRAINED LANGUAGE MODEL (PLM) COMPRESSION

We define an encoder-based PLM compression problem as follows. We have an accurate PLM $\mathcal{T}$ finetuned for the target task, which predicts the label $y$ for each instance $\boldsymbol{x}$, and a sample dataset $\mathbb{D} = \{(\boldsymbol{x}_i, y_i)\}$. We assume that PLM $\mathcal{T}$ is too large and exceeds our FLOPs budget $\tau_{\text{FLOPs}}$. Our goal is to compress the PLM $\mathcal{T}$ to a tiny model $\mathcal{S}$ to satisfy our FLOPs budget $\tau_{\text{FLOPs}}$ while maintaining its accuracy.

---

[1]percent-point

## 2.2 TRANSFORMER ARCHITECTURE

**Transformer Encoder**. In this paper, we focus on compressing the encoder-based Transformers, such as BERT (Devlin et al., 2019) and DistilBERT (Sanh et al., 2019). The encoder-based Transformers consist of two types of sublayers: multi-head attention (MHA) and feedforward network (FFN) sublayers. For a given input $\boldsymbol{X} \in \mathbb{R}^{d \times s}$ of $s$ tokens each of which is of dimension $d$, outputs of sublayers are as follows: $\mathcal{N}(\boldsymbol{X} + \mathrm{M}(\boldsymbol{X}))$ for MHA sublayers or $\mathcal{N}(\boldsymbol{X} + \mathrm{F}(\boldsymbol{X}))$ for FFN sublayers where $\mathcal{N}$ refers to layer normalization (Ba et al., 2016). The output $\mathrm{M}(\boldsymbol{X})$ of multi-head attention with $H$ attention heads is the sum of the outputs $h_i(\boldsymbol{X}) \in \mathbb{R}^{d \times s}$ of attention heads as in Equation (1) where $\boldsymbol{B}^{out} \in \mathbb{R}^{d \times s}$ is a bias. The output $h_i(\boldsymbol{X})$ of the $i$th attention head is decomposed into the output projection $\boldsymbol{W}_i^{out} \in \mathbb{R}^{d \times d_h}$ and the intermediate feature $f_i(\boldsymbol{X}) \in \mathbb{R}^{d_h \times s}$ which are the outputs of a dot-product self-attention with dimension $d_h$.

$$\mathrm{M}(\boldsymbol{X}) = \left(\sum_{i=1}^{H} h_i(\boldsymbol{X})\right) + \boldsymbol{B}^{out} \;\; where \;\; h_i(\boldsymbol{X}) = \;\; \boldsymbol{W}_i^{out} f_i(\boldsymbol{X}) \tag{1}$$

The output $\mathrm{F}(\boldsymbol{X})$ of a feedforward network with $N$ intermediate neurons is in Equation (2), where $n_i(\boldsymbol{X}) \in \mathbb{R}^{d \times s}$ is the partial output of the $i$th neuron and $\boldsymbol{C}^{out} \in \mathbb{R}^{d \times s}$ is a bias. The output $n_i(\boldsymbol{X})$ of the $i$th neuron is computed by two linear transformations and is decomposed into the output projection $\boldsymbol{v}_i^{out} \in \mathbb{R}^{d \times 1}$ and intermediate feature $g_i(\boldsymbol{X}) \in \mathbb{R}^{1 \times s}$.

$$\mathrm{F}(\boldsymbol{X}) = \left(\sum_{i=1}^{N} n_i(\boldsymbol{X})\right) + \boldsymbol{C}^{out} \;\; where \;\; n_i(\boldsymbol{X}) = \;\; \boldsymbol{v}_i^{out} g_i(\boldsymbol{X}) \tag{2}$$

**Pruning Criteria**. In this paper, we aim to identify and prune unnecessary attention heads and neurons following previous works (Michel et al., 2019; Kwon et al., 2022b). We introduce mask variables $\boldsymbol{\zeta} = [\zeta_1, \zeta_2, ..., \zeta_H]^T \in \mathbb{R}^H$ and $\boldsymbol{\xi} = [\xi_1, \xi_2, ..., \xi_N]^T \in \mathbb{R}^N$ to indicate the pruning status of attention heads and neurons, respectively; $\zeta_i = 0$ means the $i$th attention head is pruned. The masked outputs of $\mathrm{M}(\boldsymbol{X}; \boldsymbol{\zeta})$ and $\mathrm{F}(\boldsymbol{X}; \boldsymbol{\xi})$ are described in Equation (3).

$$\mathrm{M}(\boldsymbol{X}; \boldsymbol{\zeta}) = \left(\sum_{i=1}^{H} \zeta_i h_i(\boldsymbol{X})\right) + \boldsymbol{B}^{out} \;\; and \;\; \mathrm{F}(\boldsymbol{X}; \boldsymbol{\xi}) = \left(\sum_{j=1}^{N} \xi_j n_j(\boldsymbol{X})\right) + \boldsymbol{C}^{out} \tag{3}$$

All mask variables are initialized to 1, which preserves the original inference result. Once the mask variables are determined after mask search, pruning of attention heads and neurons whose mask variables are zero does not affect the inference results.

## 2.3 THE LOSS OF KNOWLEDGE AFTER PRUNING

In existing works (Hinton et al., 2015; Romero et al., 2015; Mirzadeh et al., 2020; Son et al., 2021; Kim et al., 2021a; Jang et al., 2023), large models are employed to enhance the accuracy of smaller models by transferring their knowledge, and pretrained models are widely adopted for this purpose in the context of model compression (Sun et al., 2019; Sanh et al., 2019; Jiao et al., 2020; Wang et al., 2020a; Kim et al., 2022; 2023; Cho & Kang, 2022). It is demonstrated that the knowledge of the pretrained models can be extracted from their soft label prediction and intermediate representations, and imitating them improves the generalization performance of the compressed model. For a given input $\boldsymbol{x}$, the amount $K_{pred}(\boldsymbol{x}; \boldsymbol{m})$ of the lost predictive knowledge of the compressed model $\mathcal{S}$ out of the pretrained model $\mathcal{T}$ is defined in Equation (4) (Hinton et al., 2015; Sun et al., 2019; Jiao et al., 2020) where $\boldsymbol{m} \in \mathbb{R}^{L(N+H)}$ is the pruning mask of the compressed model $\mathcal{S}$ with $L$ layers. $D_{\mathrm{KL}}$ is KL-divergence, and $\hat{\boldsymbol{z}}_{\mathcal{T}}(\boldsymbol{x}; \mathbb{1}_{|\boldsymbol{m}|})$ and $\hat{\boldsymbol{z}}_{\mathcal{S}}(\boldsymbol{x}; \boldsymbol{m})$ are logits of the pretrained and the compressed models, respectively. $s_\gamma$ is a softmax function with the temperature of $\gamma$. $\mathbb{1}_{|\boldsymbol{m}|} \in \mathbb{R}^{|\boldsymbol{m}|}$ is a vector of ones indicating an unpruned status.

$$K_{pred}(\boldsymbol{x}; \boldsymbol{m}, \gamma) = \gamma^2 D_{\mathrm{KL}}(s_\gamma(\hat{\boldsymbol{z}}_{\mathcal{T}}(\boldsymbol{x}; \mathbb{1}_{|\boldsymbol{m}|}) || s_\gamma(\hat{\boldsymbol{z}}_{\mathcal{S}}(\boldsymbol{x}; \boldsymbol{m})))) \tag{4}$$

For the $l$th sublayer, the amount $K_{rep,l}$ of lost representational knowledge regarding intermediate representations is defined in Equation (5) (Romero et al., 2015; Sun et al., 2020; Tang et al., 2019) where subscript of $\mathcal{S}$ and $\mathcal{T}$ represents the compressed model $\mathcal{S}$ and the pretrained model $\mathcal{T}$, respectively. $\boldsymbol{X}_l$ is the input of the $l$th sublayer and it is added due to the residual connection. $\mathrm{Sub}_l$ is the

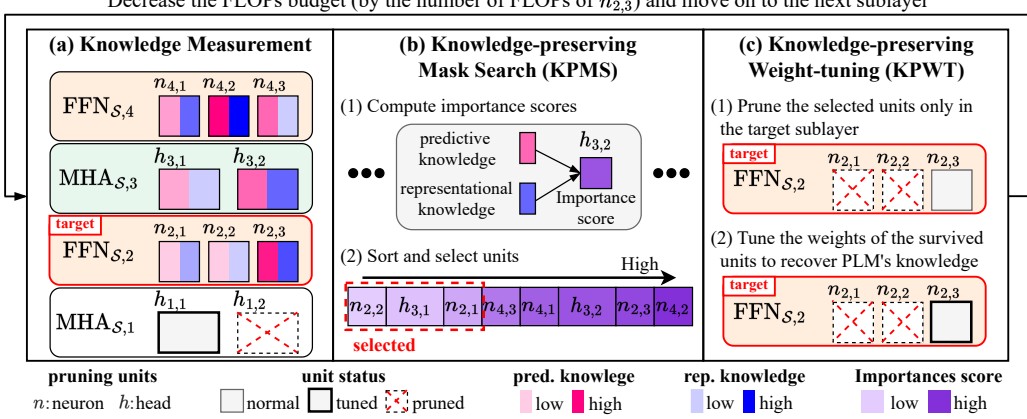

Figure 2: Illustration of K-prune when the second sublayer is our target (best viewed in color). See Section 3.1 for details.

sublayer function of the $l$th sublayer which is either $\mathrm{M}(\boldsymbol{X})$ or $\mathrm{F}(\boldsymbol{X})$, and $\boldsymbol{m}_l$ is a vector of mask variables in the $l$th sublayer of $\mathcal{S}$.

$$K_{rep,l}(\boldsymbol{X}_{\mathcal{T},l}, \boldsymbol{X}_{\mathcal{S},l}; \boldsymbol{m}_l) = \left\| \boldsymbol{X}_{\mathcal{T},l} + \mathrm{Sub}_{\mathcal{T},l}(\boldsymbol{X}_{\mathcal{T},l}; \mathbb{1}_{|\boldsymbol{m}_l|}) - \boldsymbol{X}_{\mathcal{S},l} - \mathrm{Sub}_{\mathcal{S},l}(\boldsymbol{X}_{\mathcal{S},l}; \boldsymbol{m}_l) \right\|_F^2 \quad (5)$$

It is crucial to reduce the amounts $K_{pred}$ and $K_{rep,l}$ of the lost knowledge to retain the accuracy of the pretrained model during compression.

## 3 PROPOSED METHOD

### 3.1 OVERVIEW

In this section, we propose K-prune, an accurate retraining-free pruning algorithm which preserves the knowledge of PLMs through sublayer-wise iterative pruning process. Before describing our main ideas, we summarize several challenges that must be tackled.

**C1**. **Importance criterion**. What aspects should we consider to find salient attention heads and neurons for preserving the knowledge of the PLM?

**C2**. **Identifying uninformative components**. How many attention heads and neurons should we prune in each iteration, and how can we select attention heads and neurons that minimize knowledge loss?

**C3**. **Minimizing the loss of knowledge**. Pruning induces the loss of knowledge of the PLM, leading to severe accuracy degradation. How can we efficiently recover the lost knowledge of PLM after pruning?

We address these challenges with the following main ideas.

**I1**. **Knowledge measurement** (**Section 3.2**). We gauge the amount of inherent knowledge regarding both label prediction and intermediate representations to estimate the saliency of masked units.

**I2**. **Knowledge-preserving mask search** (**Section 3.3**). In every iteration, we identify the meaningless masked units in the target sublayer considering their global importance which reflects both predictive and representational knowledge.

**I3**. **Knowledge-preserving weight-tuning** (**Section 3.4**). We remove the identified meaningless masked units only in the target sublayer and reconstruct the knowledge of the PLM through efficient weight-tuning.

K-prune iteratively performs sublayer-wise pruning, from the bottom to the top sublayer, with the following three steps: (a) knowledge measurement, (b) knowledge-preserving mask search, and (c) knowledge-preserving weight-tuning. We illustrate a pruning process for a two-layered Transformer Encoder with four sublayers when the second sublayer is our target in Figure 2. In the first step,

(a) we measure the amount of inherent predictive and representational knowledge in each masked unit (attention head and neuron) in the target sublayer and sublayers above the target sublayer (e.g., from the second to the fourth sublayers in Figure 2). The red and blue colors indicate the amounts of predictive and representational knowledge, respectively; darker colors denote richer knowledge. We measure the amount of knowledge in the above sublayers to consider the global importance of masked units in the target sublayer in step (b). We do not measure the knowledge of the sublayers (e.g., $MHA_{S,1}$) below the target sublayer since they have already been pruned. Then, (b) we compute the importance scores for each masked unit considering both predictive and representational knowledge and sort them. We select the masked units with the least importance scores to be pruned considering the FLOPs budget. The number of selected masked units in the target sublayer is determined according to the global importance of the target sublayer since we evaluate and compare the importance scores of masked units in all of the unpruned sublayers. After that, (c) we prune the selected components (e.g., $n_{2,1}$ and $n_{2,2}$) from the target sublayer and tune the weights of the remaining components (e.g., $n_{2,3}$), in the target sublayer on a small sample dataset to recover the PLM's knowledge. We decrease the FLOPs constraint by the number of FLOPs of the remaining components and then move on to the next sublayer. K-prune accurately compresses the model since it iteratively prunes a small amount of masked units in each sublayer considering their global importance after reconstructing the knowledge of the previous sublayers. The running time of K-prune is significantly low since it performs only an efficient weight-tuning on a small sample dataset. We elaborate on the details of each step in the following sections.

## 3.2 KNOWLEDGE MEASUREMENT

We use the amount of both predictive and representational knowledge in each attention head and neuron as a metric to estimate their saliency in identifying uninformative attention heads and neurons. We measure the amount of knowledge contained within each attention head and neuron by evaluating the loss of knowledge after pruning it. For $i$th pruning mask $m_{l,i}$ in the $l$th sublayer, we reformulate the functions in Equations (4) and (5), which state the amount of knowledge loss, as single-variable functions by assuming that all mask variables are independent, i.e. $K_{pred}(\boldsymbol{x}; \boldsymbol{m}, \gamma) \approx K_{pred}(\boldsymbol{x}; m_{l,i}, \gamma)$ and $K_{rep,l}(\boldsymbol{X}_{\mathcal{T},l}, \boldsymbol{X}_{\mathcal{S},l}; \boldsymbol{m}_l) \approx K_{rep,l}(\boldsymbol{X}_{\mathcal{T},l}, \boldsymbol{X}_{\mathcal{S},l}; m_{l,i})$, respectively. Then, the predictive and representational knowledge within an attention head or neuron which corresponds to the mask $m_{l,i}$ is estimated as $K_{pred}(\boldsymbol{x}; m_{l,i} = 0, \gamma)$ and $K_{rep}(\boldsymbol{X}_{\mathcal{T},l}, \boldsymbol{X}_{\mathcal{S},l}; m_{l,i} = 0)$, respectively.

We approximate the average of the amount $K_{pred}(\boldsymbol{x}; m_{l,i} = 0, \gamma)$ of predictive knowledge of $m_{l,i}$ on the sample dataset $\mathbb{D}$ as in Equation (6) by applying Taylor expansion and Fisher Information (LeCun et al., 1989; Kwon et al., 2022b).

$$\frac{1}{|\mathbb{D}|} \sum_{\boldsymbol{x} \in \mathbb{D}} K_{pred}(\boldsymbol{x}; m_{l,i} = 0, \gamma) \approx \frac{1}{|\mathbb{D}|} \sum_{\boldsymbol{x} \in \mathbb{D}} \left( \frac{1}{2\gamma^2} \left( \frac{\partial K_{pred}(\boldsymbol{x}; m_{l,i} = 1, \gamma)}{\partial m_{l,i}} \right)^2 \right) \quad (6)$$

We estimate the amount $K_{rep,l}(\boldsymbol{X}_{\mathcal{T},l}, \boldsymbol{X}_{\mathcal{S},l}; m_{l,i} = 0)$ of representational knowledge within the $i$th component in the $l$th sublayer, which corresponds to the target mask $m_{l,i}$, by the MSE loss between the outputs of the $l$th sublayers of the pretrained model $\mathcal{T}$ and the compressed model $\mathcal{S}$ as in Equation (7). We introduce a mask vector $\boldsymbol{m}_{l \setminus i} \in \mathbb{R}^{|\boldsymbol{m}_l|}$ to indicate the pruning of the $i$th masked unit, and all elements of $\boldsymbol{m}_{l \setminus i}$ are one except for the $i$th element which is zero. By assuming that the two inputs $\boldsymbol{X}_{\mathcal{T},l}$ and $\boldsymbol{X}_{\mathcal{S},l}$ are the same, $K_{rep,l}(\boldsymbol{X}_{\mathcal{T},l}, \boldsymbol{X}_{\mathcal{S},l}; m_{l,i} = 0)$ becomes the norm of the output of the components as in Equation (8) since the masked outputs of sublayers are computed as the sum of unpruned attention heads or neurons as in Equation (3).

$$K_{rep,l}(\boldsymbol{X}_{\mathcal{T},l}, \boldsymbol{X}_{\mathcal{S},l}; m_{l,i} = 0) = \left\| \boldsymbol{X}_{\mathcal{T},l} + \text{Sub}_{\mathcal{T},l}(\boldsymbol{X}_{\mathcal{T},l}, \mathbb{1}_{\boldsymbol{m}_l}) - \boldsymbol{X}_{\mathcal{S},l} - \text{Sub}_{\mathcal{S},l}(\boldsymbol{X}_{\mathcal{S},l}, \boldsymbol{m}_{l \setminus i}) \right\|_F^2 \quad (7)$$

$$\approx \begin{cases} \|h_{l,i}(X_{\mathcal{S},l})\|_F^2 & \text{for MHA sublayers} \\ \|n_{l,i}(X_{\mathcal{S},l})\|_F^2 & \text{for FFN sublayers} \end{cases} \quad (8)$$

## 3.3 KNOWLEDGE-PRESERVING MASK SEARCH (KPMS)

We propose Knowledge-preserving mask search (KPMS), an accurate mask search algorithm which finds an accurate non-uniform pruning mask for each sublayer. In KPMS, we estimate the importance of each masked unit using the amount of knowledge in the masked unit to minimize the knowledge loss after pruning. We estimate the importance score not only in the target sublayer but also in the sublayers above the target sublayer to control the number of masked units to prune in the target

---

**Algorithm 1** Knowledge-Preserving Mask Search (KPMS)

---

**Input** : Sample dataset $\mathbb{D}$, pretrained model $\mathcal{T}$, compressed model $\mathcal{S}$,
    FLOPs $F_h$ and $F_n$ of a head and a neuron, FLOPs budget $\tau_{\text{FLOPs}}$,
    temperature $\gamma$ for Equation (4) and balance coefficients $\lambda$ and $\mu$ for Equation (9).
**Output** : the sets $\mathbb{P}_{head}$ and $\mathbb{P}_{neuron}$ of attention heads and neurons to be pruned

---

1: $\boldsymbol{K}_{head}^{pred}, \boldsymbol{K}_{head}^{rep}, \boldsymbol{K}_{neuron}^{pred}, \boldsymbol{K}_{neuron}^{rep} \leftarrow$ measure-knowledge$(\mathcal{S}, \mathcal{T}, \mathbb{D}, \gamma)$     $\triangleright$ Equations (6), (8)
2: $\boldsymbol{Z}_{head}, \boldsymbol{Z}_{neuron} \leftarrow$ scoring$(\boldsymbol{K}_{head}^{pred}, \boldsymbol{K}_{head}^{rep}, \boldsymbol{K}_{neuron}^{pred}, \boldsymbol{K}_{neuron}^{rep}, \mu, \lambda, F_h, F_n)$    $\triangleright$ Equation (9)
3: $\tilde{\boldsymbol{Z}} \leftarrow$ concat-and-sort-ascending-order$(\boldsymbol{Z}_{head}, \boldsymbol{Z}_{neuron})$
4: $p \leftarrow 0, f \leftarrow |\boldsymbol{Z}_{head}|F_h + |\boldsymbol{Z}_{neuron}|F_n$
5: **while** $f > \tau_{\text{FLOPs}}$ **do**
6:     $\nu \leftarrow \tilde{\boldsymbol{Z}}[p]$                                    $\triangleright$ candidate threshold
7:     $n_h \leftarrow |\{h|S_{head}[h] \geq \nu\}|, n_n \leftarrow |\{n|S_{neuron}[n] \geq \nu\}|$    $\triangleright$ remained heads and neurons
8:     $p \leftarrow p + 1, f \leftarrow n_h F_h + n_n F_n$               $\triangleright$ FLOPs of the compressed model
9: **end while**
10: $\nu^* \leftarrow \nu$
11: $\mathbb{P}_{head} \leftarrow \{h|\boldsymbol{Z}_{head}[h] < \nu^*\}, \mathbb{P}_{neuron} \leftarrow \{n|\boldsymbol{Z}_{neuron}[n] < \nu^*\}$        $\triangleright$ selected to be pruned

---

sublayer, considering their global importance. KPMS is described in Algorithm 1. We begin KPMS by measuring the amount of knowledge in attention heads and neurons to estimate their importance score (line 1). We evaluate the amount of both predictive and representational knowledge in attention heads and neurons on the sample dataset $\mathbb{D}$ following Equations (6) and (8). $\boldsymbol{K}_{head}^{pred}, \boldsymbol{K}_{head}^{rep} \in \mathbb{R}^{LH}$ and $\boldsymbol{K}_{neuron}^{pred}, \boldsymbol{K}_{neuron}^{rep} \in \mathbb{R}^{LN}$ in Algorithm 1 represent the vectors of the estimated amount of knowledge in all attention heads and neurons in the model, respectively, where $L$ is the number of layers, i.e. there are $L$ MHA sublayers and $L$ FFN sublayers. In detail, we set the amount of knowledge as 0 for the attention heads and neurons in the sublayers below the target sublayer, which was pruned in previous steps, in order to ignore them during the mask search. Then, we estimate the importance score of each attention head and neuron as the weighted sum of the amount of the predictive and representational knowledge with a balance coefficient $\lambda$ as in Equations (9) (line 2). We divide the scores of attention heads and neurons by their number of FLOPs ($F_h$ for attention heads and $F_n$ for neurons) in order to consider the amount of importance score per FLOP. We multiply the scores $\boldsymbol{Z}_{head}$ of attention heads by another balance coefficient $\mu$ to reflect the different sensitivity between attention heads and neurons.

$$\boldsymbol{Z}_{head} = \mu \left(\boldsymbol{K}_{head}^{pred} + \lambda \boldsymbol{K}_{head}^{rep}\right)/F_h \;\; and \;\; \boldsymbol{Z}_{neuron} = \left(\boldsymbol{K}_{neuron}^{pred} + \lambda \boldsymbol{K}_{neuron}^{rep}\right)/F_n \quad (9)$$

We concatenate two vectors $\boldsymbol{Z}_{neuron}$ and $\boldsymbol{Z}_{head}$, and then sort the concatenated vector in increasing order to find the threshold for pruning (line 3). We sequentially obtain threshold candidates $\nu$ from the sorted score vector $\tilde{\boldsymbol{Z}}$ until the FLOPs $f$ of the compressed model pruned by the threshold $\nu$ is smaller than our FLOPs budget $\tau_{\text{FLOPs}}$ (lines 4-9). Consequently, we get the optimal threshold $\nu^*$, and find the sets $\mathbb{P}_{head}$ and $\mathbb{P}_{neuron}$ containing the indices of heads and neurons whose importance score is lower than $\nu^*$, respectively (lines 10-11).

## 3.4 KNOWLEDGE-PRESERVING WEIGHT-TUNING (KPWT)

We propose Knowledge-preserving weight-tuning (KPWT), an efficient weight-tuning process that reconstructs the distorted knowledge of the PLM after pruning. In every sublayer-wise iteration of K-prune, we prune only masked units in the target sublayer to formulate the problem of knowledge reconstruction as a problem which requires an extremely short time to solve. When the $l$th sublayer is our target, we prune masked units in the $l$th sublayer if they are included in $\mathbb{P}_{head}$ or $\mathbb{P}_{neuron}$ of KPMS. Then, we formulate the knowledge reconstructing problem as the problem of minimizing the loss $K_{rep,l}(\boldsymbol{X}_{\mathcal{T},l}, \boldsymbol{X}_{\mathcal{S},l}; \boldsymbol{m}_l)$ of representational knowledge of the $l$th sublayer in Equation (5). Equation (10) is the reformulated problem of Equation (5) for MHA sublayers where $\zeta_{l,i}^*$ represents the found mask of the $i$th attention head in the $l$th sublayer in Algorithm 1, i.e. the value of mask $\zeta_{l,i}^*$ is 0 if the index $(lH + i)$ of its corresponding attention head is in $\mathbb{P}_{head}$ or 1 otherwise. We modify the problem as the linear least square problem over the set of output projections $\{\boldsymbol{W}_{l,i}^{out}\}_{i=1}^{H}$ to exploit the efficiency of the linear solver. We collect the sublayer outputs $(\boldsymbol{X}_{\mathcal{T},l} + \mathrm{M}_{\mathcal{T},l}(\boldsymbol{X}_{\mathcal{T},l}, \mathbb{1}))$ of the pretrained model, which does not change during a pruning process, at the first iteration of

K-prune and reuse them for every iteration. We collect the set $\{f_{l,i}(\boldsymbol{X}_{\mathcal{S},l})\}_{i=1}^{H}$ of features when we measure the knowledge in KPMS (line 1 in Algorithm 1) Analogously, we formulate the problem for FFN sublayers as in Equation (11) where $\xi_{l,i}^*$ represents the found mask of the $i$th neuron of the $l$th sublayer. The subscript $l$ in a symbol represents that the symbol is related to the $l$th sublayer. We tune weights ($\boldsymbol{W}_{l,i}^{out}$ or $\boldsymbol{v}_{l,i}^{out}$) to achieve high accuracy even at high compression rates.

$$\underset{\{\boldsymbol{W}_{l,i}^{out}\}_{i=1}^{H}}{\arg\min} \left\| \boldsymbol{X}_{\mathcal{T},l} + \mathrm{M}_{\mathcal{T},l}(\boldsymbol{X}_{\mathcal{T},l}, \mathbb{1}_H) - \boldsymbol{X}_{\mathcal{S},l} - \left( \sum_{i=1}^{H} \zeta_{l,i}^* \boldsymbol{W}_{l,i}^{out} f_{l,i}(\boldsymbol{X}_{\mathcal{S},l}) \right) - \boldsymbol{B}_l^{out} \right\|_F^2 \tag{10}$$

$$\underset{\{\boldsymbol{v}_{l,i}^{out}\}_{i=1}^{N}}{\arg\min} \left\| \boldsymbol{X}_{\mathcal{T},l} + \mathrm{F}_{\mathcal{T},l}(\boldsymbol{X}_{\mathcal{T},l}, \mathbb{1}_N) - \boldsymbol{X}_{\mathcal{S},l} - \left( \sum_{i=1}^{N} \xi_{l,i}^* \boldsymbol{v}_{l,i}^{out} g_{l,i}(\boldsymbol{X}_{\mathcal{S},l}) \right) - \boldsymbol{C}_l^{out} \right\|_F^2 \tag{11}$$

We use a linear solver[2] in PyTorch (Paszke et al., 2019) to solve Equations (10) and (11). Note that the time for solving the problems in Equations (10) and (11) is shorter than a second in a typical desktop computer, which is several magnitudes smaller than those of conventional retraining processes in existing works (Xia et al., 2022; Hou et al., 2020; Lagunas et al., 2021; Liu et al., 2021), and does not require any hyperparameter tuning. After pruning and knowledge reconstruction, we decrease our FLOPs constraint by the FLOPs of the remaining attention heads or neurons in the $l$th sublayer. Then, we move on to the $(l+1)$th sublayer with the adjusted FLOPs constraint. This enables K-prune to satisfy the FLOPs constraint through a single run without any interventions from users.

## 4 EXPERIMENTS

We perform experiments to answer the following questions about K-prune:

**Q1**. **Accuracy** (**Section 4.2**). How accurate are the models compressed with K-prune compared to the models compressed with existing retraining-free pruning algorithms?

**Q2**. **Inference speed** (**Section 4.3**). How fast are the models compressed with K-prune compared to the models compressed with existing retraining-free pruning algorithms?

**Q3**. **Efficiency** (**Section 4.4**). How efficient is K-prune compared to the existing pruning algorithms including retraining-based ones in terms of both accuracy and pruning cost?

**Q4**. **Ablation study** (**Section 4.5**). Do our ideas of K-prune, i.e. knowledge-based importance criteria, KPMS, and KPWT, improve the accuracy of the compressed models?

### 4.1 EXPERIMENTAL SETUP

**Setup**. We use PyTorch (Paszke et al., 2019), and the weights of the pretrained models in Transformers (Wolf et al., 2020). We evaluate the performance of compressing the pretrained BERT (Devlin et al., 2019) and DistilBERT (Sanh et al., 2019) models on GLUE (Wang et al., 2019), SQuAD v1.1 (Rajpurkar et al., 2016), and v2 (Rajpurkar et al., 2018) under diverse compression rates. We use FLOPs as a compression measure which is computed on the average sequence length of each dataset. We report the compression rate as ratio of the removed number of FLOPs after pruning. We use NVIDIA 1080 Ti for all experiments.

**Hyperparameters**. We use 100K tokens from the training dataset as a sample dataset, and exploit the pretrained tokenizers in Transformers (Wolf et al., 2020) for counting. The size of the sample dataset is small compared to the GLUE and SQuAD datasets, e.g. around 0.64% of MNLI (Williams et al., 2018) dataset. We fix random seeds from 0 to 4 and report the average performance of the 5 runs. We use two combinations of hyperparameters $(\gamma, \lambda, \mu) \in \{(2, 0, 64), (2, 0.00025, 64)\}$ for all experiments of K-prune.

**Competitors**. We compare the performance of K-prune with existing retraining-free pruning algorithms for PLMs: Kwon et al. (2022b) and KCM (Nova et al., 2023). We compare the pruning efficiency with state-of-the-art retraining-based pruning algorithms for PLMs, DynaBERT (Hou et al., 2020) and EBERT (Liu et al., 2021) which show the best tradeoff in terms of accuracy vs. pruning cost, outperforming FLOP (Wang et al., 2020b), Sajjad et al. (2023), CoFi (Xia et al., 2022), and BMP (Lagunas et al., 2021) as reported in Kwon et al. (2022b). We use entire datasets for training retraining-based algorithms.

---

[2]torch.linalg.lstsq

Table 1: Comparison of inference speed of the models compressed by K-prune and competitors. We report the best result of the compressed models whose accuracy degradation is lower than 3%p. K-prune shows the highest acceleration, giving up to 2.93× faster speed than the uncompressed model.

| Method | MRPC | STS-B | SQuAD$_{1.1}$ | SQuAD$_{2.0}$ | Avg.[*] |
|---|---|---|---|---|---|
| KCM (Nova et al., 2023) | 1.08× | 1.23× | 1.20× | 1.08× | 1.15× |
| Kwon et al. (2022b) | 1.59× | 2.10× | 2.09× | 1.75× | 1.87× |
| K-prune (ours) | 2.66× | 2.43× | 2.60× | 2.93× | 2.65× |

[*] Geometric mean

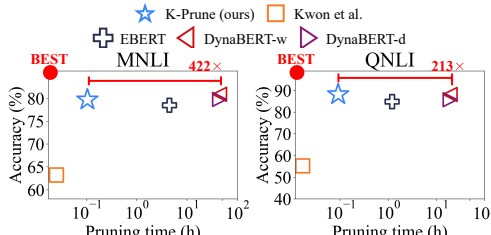

Table 2: Evaluation of K-prune and its variants under a compression rate of 80%. Each of the proposed ideas successfully improves the accuracy of the compressed models, and K-prune shows the best results. We get the largest accuracy improvement from KPWT.

Figure 3: Accuracy of compressed models vs. time cost for pruning under a compression rate of 75%. K-prune (blue star) shows the best trade-off among both retraining-free and retraining-based pruning algorithms.

| Method | MRPC | SQuAD[*] |
|---|---|---|
| K-prune | 84.80 | 74.16 |
| K-prune - $K_{pred}, K_{rep}$ | 84.07 | 72.55 |
| K-prune - KPMS | 81.71 | 67.10 |
| K-prune - KPWT | 68.38 | 16.50 |

[*] SQuAD$_{1.1}$

## 4.2 ACCURACY OF THE COMPRESSED MODELS (Q1)

Figure 1 shows a comparison of the accuracy vs. reduced FLOPs of the compressed models generated by K-prune and competitors on diverse tasks and models. The black dotted line indicates the 3%p accuracy degradation from the baseline models. In all settings, K-prune outperforms all competitors in large gaps by up to 58%p. The accuracy gap between K-prune and the competitors grows larger as the compression ratio gets higher since their one-shot pruning process fails to cope with the pruning errors; especially, KCM shows drastic accuracy degradation as the ratio of reduced FLOPs increases since it cannot prune attention heads. Our results demonstrate that K-prune effectively addresses the significant accuracy degradation problem by preserving the knowledge of PLM via a thoughtfully designed iterative pruning process incorporating our novel ideas: KPMS and KPWT.

## 4.3 ACCELERATION ON COMMODITY HARDWARE (Q2)

We compare the inference speed of the compressed models whose accuracy drop is lower than 3%p compared to the baseline model. We use randomly generated input sequences whose length is equal to the average length of input sequences in each task. We use a batch size of 32 for all experiments. We summarize the highest acceleration ratio of K-prune and competitors compared to the baseline model in Table 1. K-prune consistently shows the highest acceleration compared to existing methods on all tasks. K-prune achieves up to 2.93× faster inference speed compared to the baseline model on commodity hardware, while other methods achieve at most 2.10× faster inference speed.

## 4.4 COMPARISON WITH RETRAINING-BASED PRUNING ALGORITHMS (Q3)

In Sections 4.2 and 4.3, we demonstrate that K-prune outperforms existing retraining-free algorithms with large margins in terms of both accuracy and inference speed. In this section, we compare K-prune with both retraining-free and retraining-based pruning algorithms to show the efficiency of K-prune. We compare the cost of each pruning algorithm by measuring the time for pruning in hours, and the accuracy of the compressed models for 75% compression rate on MNLI and QNLI datasets in Figure 3. DynaBERT-d and DynaBERT-w are two variants of DyanaBERT with and without applying depth multipliers, respectively. Note that K-prune shows comparable or better accuracy in all settings compared to Kwon et al., EBERT, DynaBERT-w, and DynaBERT-d while showing up to 422 × lower pruning cost. Thus, K-prune shows the best trade-off regarding the accuracy and pruning time among both the retraining-based and retraining-free pruning algorithms.

## 4.5 ABLATION STUDY (Q4)

We perform an ablation study to show that each technique of K-prune, such as knowledge-based importance criteria, KPMS, and KPWT, improves the accuracy of the compressed model. We summarize the results in Table 2 under the compression rate of 80% on MRPC and SQuAD$_{1.1}$. Each row of Table 2 depicts the change of performance when an individual idea is omitted from K-prune. -$K_{pred}$, $K_{rep}$ shows the results from using the magnitude of the derivative of cross entropy instead of the knowledge-based importance criterion, -KPMS denotes cases where pruning is performed uniformly across sub-layers without considering global importance, and -KPWT represents that iterative pruning and weight-tuning are not employed. Our results show that all ideas contribute to the performance enhancement, and KPWT shows the most significant impact.

## 5 RELATED WORKS

### 5.1 TRANSFORMER COMPRESSION

Transformer compression algorithms are designed to reduce the size and inference time of Transformer. These algorithms are categorized based on the aspects they focus on: quantization (Kim et al., 2021b; Piao et al., 2022; Kwon et al., 2022a), low-rank approximation (Wang et al., 2022; Cordonnier et al., 2020), parameter sharing (Lan et al., 2020; Jeon et al., 2023), structured pruning (Hou et al., 2020; Liu et al., 2021; Kwon et al., 2022b; Nova et al., 2023), and unstructured pruning (Sanh et al., 2020; Yao et al., 2021). In this paper, we focus on structured pruning which guarantees instant acceleration on commodity hardware. Note that other types of algorithms are complementary to structured pruning in achieving a higher compression rate, as they address different kinds of inefficiencies (Lazarevich et al., 2021; Frantar & Alistarh, 2023).

### 5.2 STRUCTURED PRUNING FOR TRANSFORMERS

Structured pruning algorithms for Transformers are divided into two groups: retraining-based and retraining-free ones. Earlier approaches for structured pruning (Hou et al., 2020; Liu et al., 2021; Lagunas et al., 2021; Xia et al., 2022) are retraining-based algorithms which generate highly sparse and accurate models based on their sophisticated training using entire datasets. However, these algorithms demand extensive retraining costs and intensive hyperparameter tuning, limiting their usage; for large language models (Brown et al., 2020; Zhang et al., 2022), retraining-based algorithms are intractable. For example, DynaBERT (Hou et al., 2020) requires three individual retraining processes for pruning BERT. Retraining-free algorithms (Kwon et al., 2022b; Nova et al., 2023) are proposed to reduce the expensive pruning cost by removing retraining processes. However, they face a significant accuracy drop since they fail to cope with pruning errors. Our proposed K-prune resolves the accuracy degradation problem, achieving both speed and accuracy.

## 6 CONCLUSION

We propose K-prune, an accurate retraining-free structured pruning algorithm for encoder-based PLMs. We address the problem of severe accuracy degradation in prior retraining-free pruning algorithms by carefully designing an iterative pruning algorithm to preserve the knowledge of PLMs. K-prune achieves remarkable accuracy improvement up to 58.02%p better performance than existing retraining-free pruning algorithms. Future works include extending our method for decoder-based models.

**Acknowledgments**. This work was supported by Youlchon Foundation. This work was also supported by Institute of Information & communications Technology Planning & Evaluation(IITP) grant funded by the Korea government(MSIT) [No.2020-0-00894, Flexible and Efficient Model Compression Method for Various Applications and Environments], [No.2021-0-01343, Artificial Intelligence Graduate School Program (Seoul National University)], and [NO.2021-0-02068, Artificial Intelligence Innovation Hub (Artificial Intelligence Institute, Seoul National University)]. The Institute of Engineering Research at Seoul National University provided research facilities for this work. The ICT at Seoul National University provides research facilities for this study. U Kang is the corresponding author.

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

# A   SYMBOLS AND DEFINITIONS

We summarize the definitions of the symbols in Table 3. For simplicity, we omit the notation $l$ representing the $l$th sub-layer if omiting $l$ does not introduce any confusion.

Table 3: Symbols and descriptions.

| Symbol | Description |
|---|---|
| $\mathcal{T}, \mathcal{S}$ | pre-trained and compressed models |
| $\text{Sub}(\cdot)$ | a sub-layer function |
| $\text{M}(\cdot), \text{F}(\cdot)$ | sub-layer functions for MHA and FFN sub-layers |
| $h(\cdot), n(\cdot)$ | an attention head and a neuron in an intermediate layer |
| $f(\cdot), g(\cdot)$ | intermediate features of an attention head and a neuron |
| $\boldsymbol{W}^{out}, \boldsymbol{v}^{out}$ | output projections for an attention head and a neuron |
| $\boldsymbol{B}^{out}, \boldsymbol{C}^{out}$ | biases for output projections in MHA and FFN sub-layers |
| $\zeta, \xi$ | masks for an attention head and a neuron |
| $\mathbb{1}_d$ | a length $d$ vector filled with ones |
| $\boldsymbol{m}_{l \backslash i}$ | a mask vector filled with ones except the $i$th element which is zero |
| $H$ | the number of attention heads in an MHA sub-layer |
| $N$ | the number of neurons in a FFN sub-layer |
| $d$ | the dimension of token embeddings |
| $s$ | a sequence length |
| $d_h$ | the dimension of projected embeddings in attention heads |
| $\mathbb{D}$ | a sample dataset |
| $(\boldsymbol{x}, y)$ | a tuple of a data point and its label in $\mathbb{D}$ |
| $\boldsymbol{X}$ | an input of a sub-layer |
| $K_{pred}, K_{rep}$ | predictive and representational knowledge |
| $Z_{head}, Z_{neuron}$ | importance scores of attention heads and neurons |
| $\gamma$ | the temperature of softmax functions |
| $\lambda$ | a coefficient for balancing $K_{pred}$ and $K_{rep}$ |
| $\mu$ | a coefficient for balancing $S_{head}$ and $S_{neuron}$ |
| $\tau_{FLOPs}$ | a FLOPs constraint |
| $\text{FLOPs}(\cdot)$ | a function for measuring FLOPs of the model |
| $F_h$ | the number of FLOPs for computing the output of an attention head |
| $F_n$ | the number of FLOPs for computing the output of a neuron |

# B DERIVATIONS

We provide additional derivations for completeness.

## B.1 DERIVATION OF EQUATION (8) IN SECTION 3.2

For MHA sublayers, we derive $K_{rep,l}(\boldsymbol{X}_{\mathcal{T},l}, \boldsymbol{X}_{\mathcal{S},l}; m_{l,i} = 0) \approx \|h_{l,i}(\boldsymbol{X}_{\mathcal{S},l})\|_F^2$ as follows under the same assumption. We assume that $\boldsymbol{X}_{\mathcal{T},l} \approx \boldsymbol{X}_{\mathcal{S},l}$ since we reconstruct the output of the previous sublayer.

$$
\begin{aligned}
K_{rep,l}(\boldsymbol{X}_{\mathcal{T},l}, \boldsymbol{X}_{\mathcal{S},l}; m_{l,i} = 0) &= \left\| \boldsymbol{X}_{\mathcal{T},l} + \mathrm{Sub}_{\mathcal{T},l}(\boldsymbol{X}_{\mathcal{T},l}, \mathbb{1}_{\boldsymbol{m}_l}) - \boldsymbol{X}_{\mathcal{S},l} - \mathrm{Sub}_{\mathcal{S},l}(\boldsymbol{X}_{\mathcal{S},l}, \boldsymbol{m}_{l\backslash i}) \right\|_F^2 \\
&\approx \left\| \sum_{j=1}^H h_{l,j}(\boldsymbol{X}_{\mathcal{S},l}) + \boldsymbol{B}_l^{out} - \left( \sum_{j=1}^H h_{l,j}(\boldsymbol{X}_{\mathcal{S},l}) + \boldsymbol{B}_l^{out} - h_{l,i}(\boldsymbol{X}_{\mathcal{S},l}) \right) \right\|_F^2 \\
&= \|h_{l,i}(\boldsymbol{X}_{\mathcal{S},l})\|_F^2
\end{aligned}
$$

Analogously, we derive $K_{rep,l}(\boldsymbol{X}_{\mathcal{T},l}, \boldsymbol{X}_{\mathcal{S},l}; m_{l,i} = 0) \approx \|n_{l,i}(X_{\mathcal{S},l})\|_F^2$ for FFN sublayers as follows.

$$
\begin{aligned}
K_{rep,l}(\boldsymbol{X}_{\mathcal{T},l}, \boldsymbol{X}_{\mathcal{S},l}; m_{l,i} = 0) &= \left\| \boldsymbol{X}_{\mathcal{T},l} + \mathrm{Sub}_{\mathcal{T},l}(\boldsymbol{X}_{\mathcal{T},l}, \mathbb{1}_{\boldsymbol{m}_l}) - \boldsymbol{X}_{\mathcal{S},l} - \mathrm{Sub}_{\mathcal{S},l}(\boldsymbol{X}_{\mathcal{S},l}, \boldsymbol{m}_{l\backslash i}) \right\|_F^2 \\
&\approx \left\| \sum_{j=1}^N n_{l,j}(\boldsymbol{X}_{\mathcal{S},l}) + \boldsymbol{C}_l^{out} - \left( \sum_{j=1}^N n_{l,j}(\boldsymbol{X}_{\mathcal{S},l}) + \boldsymbol{C}_l^{out} - n_{l,i}(\boldsymbol{X}_{\mathcal{S},l}) \right) \right\|_F^2 \\
&= \|n_{l,i}(\boldsymbol{X}_{\mathcal{S},l})\|_F^2
\end{aligned}
$$

## B.2 REFORMULATED PROBLEM OF KNOWELDGE RECONSTRUCTION FOR LINEAR SOLVERS (SECTION 4.4)

We reformulate Equations (10) and ( 11) in our main text as a form of linear least square problem to use linear solvers, such as torch.linalg.lstsq, as in Equation (12).

$$
\boldsymbol{W}^* = \arg\min_{\boldsymbol{W}} \|\boldsymbol{P}\boldsymbol{W} - \boldsymbol{Q}\|_F^2 \tag{12}
$$

We derive $\boldsymbol{P}$, $\boldsymbol{W}$, and $\boldsymbol{Q}$ for MHA sub-layers as in Equation (13) where $\|$ is columnwise concatenation. $\boldsymbol{P}$ is a transpose of concatenated feature matrix of the remained attention heads after pruning and $\boldsymbol{W}$ is a concatenation of transposed weight matrices of the output projections in the remained attention heads after pruning.

$$
\begin{aligned}
\boldsymbol{P} &= \left( \|_{i \in \{i | \zeta_i \neq 0\}} f_i(\boldsymbol{X}_{\mathcal{S}}) \right)^T \\
\boldsymbol{W} &= \|_{i \in \{i | \zeta_i \neq 0\}} \left( \boldsymbol{W}_i^{out} \right)^T \\
\boldsymbol{Q} &= \left( \boldsymbol{X}_{\mathcal{T}} + \mathrm{M}_{\mathcal{T}}(\boldsymbol{X}_{\mathcal{T}}, \mathbb{1}_H) - \boldsymbol{X}_{\mathcal{S}} - \boldsymbol{B}^{out} \right)^T
\end{aligned} \tag{13}
$$

We derive $\boldsymbol{P}$, $\boldsymbol{W}$, and $\boldsymbol{Q}$ for FFN sub-layers of Equation (14) in the same logic as the MHA sub-layers.

$$
\begin{aligned}
\boldsymbol{P} &= \left( \|_{i \in \{i | \xi_i \neq 0\}} g_i(\boldsymbol{X}_{\mathcal{S}}) \right)^T \\
\boldsymbol{W} &= \|_{i \in \{i | \xi_i \neq 0\}} \left( \boldsymbol{v}_i^{out} \right)^T \\
\boldsymbol{Q} &= \left( \boldsymbol{X}_{\mathcal{T}} + \mathrm{F}_{\mathcal{T}}(\boldsymbol{X}_{\mathcal{T}}, \mathbb{1}_N) - \boldsymbol{X}_{\mathcal{S}} - \boldsymbol{C}^{out} \right)^T
\end{aligned} \tag{14}
$$

## C  DETAILED EXPERIMENTAL SETTINGS

### C.1  DATA DESCRIPTION

We summarize the characteristics of GLUE and SQuAD benchmarks in Table 4.

Table 4: Summarization of benchmark datasets.

| Name | Samples | Tokens | Task | Metric |
|------|---------|--------|------|--------|
| MRPC | 3.7k | 195k | paraphrase | accuracy |
| QQP | 364k | 11,123k | paraphrase | accuracy |
| SST-2 | 67k | 897k | sentiment | accuracy |
| STS-B | 7k | 160k | sentence similarity | Spearman corr. |
| MNLI | 393k | 15,629k | NLI* | accuracy |
| QNLI | 105k | 5,176k | QA**/ NLI | accuracy |
| $SQuAD_{1.1}$ | 88k | 15,116k | QA | F1 score |
| $SQuAD_{2.0}$ | 132k | 22,455k | QA | F1 score |

∗ natural language inference    ∗∗ question answering

### C.2  FINE-TUNING OF PLMS

We fine-tune BERT (Devlin et al., 2019) following a standard training recipe. We use fine-tuned checkpoints of DistilBERT in the github[3]. We summarize the performance of fine-tuned BERT and DistilBERT in Table 5.

Table 5: Accuracy of the fine-tuned BERT and DistilBERT.

| | MRPC | QQP | SST-2 | STS-B | MNLI | QNLI | $SQuAD_{1.1}$ | $SQuAD_{2.0}$ |
|---|------|-----|-------|-------|------|------|---------------|---------------|
| BERT | 87.01 | 91.54 | 93.12 | 89.08 | 84.90 | 91.87 | 88.51 | 76.54 |
| DistilBERT | 84.80 | 89.99 | 91.39 | 86.12 | 82.10 | 88.55 | 85.73 | 68.84 |

### C.3  TRAINING DETAILS OF K-PRUNE

**Code**. We attach our implementation of K-prune in the supplementary material. We attach scripts and detailed instructions for reproducing our experimental results.

**Hyperparameter**. In addition to the hyperparameter settings $\{(2, 1, 64), (2, 0.00025, 64)\}$ used in the main text, we provide additional results with a wider range of hyperparameter settings. We perform experiments on $SQuAD_{1.1}$ under compression rates of 40%, 60%, and 80%.

**Sensitivity analysis regarding** $\gamma$. Figure 4 shows the change of the F1 score of the model with regard to the change of the temperature $\gamma$ for softmax functions. We use $\gamma \in \{0.5, 1.0, 1.5, ..., 4.0\}$ where a higher $\gamma$ represents a smoother prediction after softmax. The F1 score of the compressed model is weakly sensitive to the change of $\gamma$. We get an accurate compressed model with $\gamma = 2$ which is used for comparison with existing works in the main text, and we get additional accuracy improvement when we use $\gamma = 1.5$.

**Sensitivity analysis regarding** $\lambda$. Figure 5 shows the change of the F1 score of the model with regard to the change of the balance coefficient $\lambda$ for representational knowledge. We use $\lambda \in \{0.25, 0.025, ..., 0.0000025\}$ where a higher $\lambda$ imposes higher importance on representational knowledge than predictive knowledge. We additionally depict the results of two cases that use only predictive or representational knowledge with the leftmost and rightmost stars in each figure. Overall, predictive knowledge plays an important role and shows higher f1 scores than representational knowledge. However, when it comes to the high compression rate, i.e. 80%, we find that using representational knowledge improves the performance of the compressed model compared to the case

---

[3]https://github.com/WoosukKwon/retraining-free-pruning

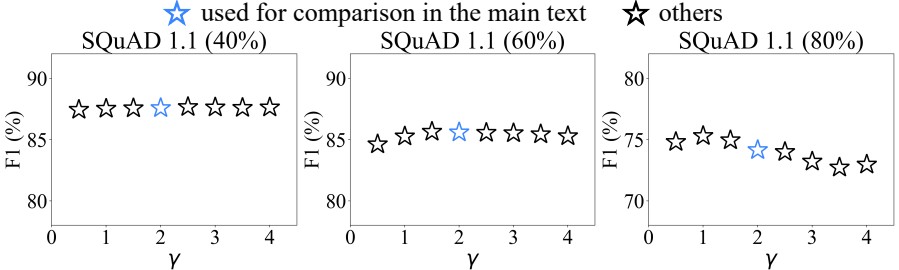

Figure 4: Change of f1 scores with regard to the change of the temperature $\gamma$ on $\text{SQuAD}_{1.1}$ under compression rates of 40%, 60%, and 80%. The f1 scores of the compressed model exhibit weak sensitivity to the alteration in $\gamma$.

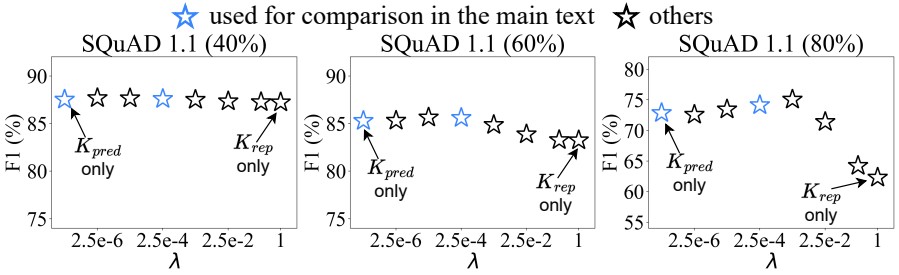

Figure 5: Change of f1 scores with regard to the change of the balance coefficient $\lambda$ on $\text{SQuAD}_{1.1}$ under compression rates of 40%, 60%, and 80%. The leftmost and rightmost stars represent the cases that use only predictive or representational knowledge, respectively. Representational knowledge is not effective by itself in general, however, it improves the accuracy of the compressed model when combined with predictive knowledge.

in which we use only predictive knowledge. We get an accurate model with $\lambda \in \{0, 0.00025\}$ which is used for comparison with existing works in the main text. We get additional accuracy improvement when we use $\lambda = 0.0025$ at the compression rate of 80%.

**Sensitivity analysis regarding** $\mu$. Figure 6 shows the change of the F1 score of the model with regard to the change of the balance coefficient $\mu$ for scores of attention heads. We use $\mu \in \{1, 2, 4, 8, ..., 2048\}$ where a higher $\mu$ imposes higher importance on the scores of the attention heads than neurons, and encourages the pruning of neurons. As a result, we find that $\mu \in [32, 128]$ consistently shows accurate results on all compression rates, and too-low or too-high value of $\mu$ shows severe accuracy degradation. We conjecture that this accuracy degradation comes from the imbalance of pruning of attention heads and neurons. We recommend using $\mu = 64$ which consistently shows accurate results.

## C.4 TRAINING DETAILS OF COMPETITORS

We summarize the training details of competitors.

### C.4.1 KWON ET AL. (2022B)

**Code**. We use the code implemented by authors in github[4].

**Hyperparameters** We use damp $= 1$ for LSMR solver[5] in CuPy and acceptable range of tuned varialbes as $[-10, 10]$ following the original paper (Kwon et al., 2022b).

---

[4]https://github.com/WoosukKwon/retraining-free-pruning
[5]cupyx.scipy.sparse.linalg.lsmr

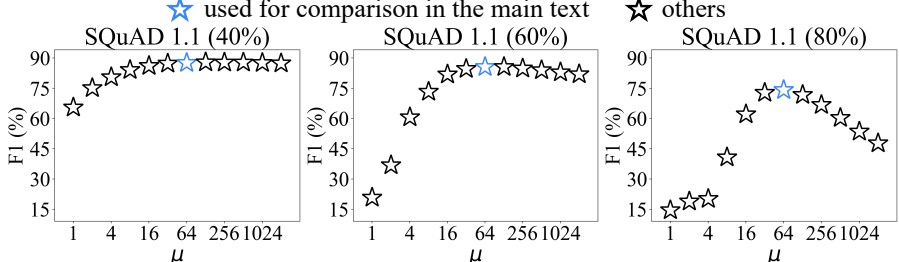

Figure 6: Change of f1 scores with regard to the change of the balance coefficient $\mu$ on SQuAD$_{1.1}$ under compression rates of 40%, 60%, and 80%. The results with $\mu \in [32, 128]$ are accurate in all settings, and too-low or too-high value of $\mu$ shows severe performance degradation.

### C.4.2   KCM (NOVA ET AL., 2023)

**Code**. We reimplement the KCM since there is no public implementation of authors.

**Hyperparameters** We use width $\sigma = 1$ of the Gaussian kernel and convergence rate $\alpha = 0.01$ as in the original paper (Nova et al., 2023). We use Min-Max normalization for normalizing $D2$ scores.

### C.4.3   DYNABERT (HOU ET AL., 2020)

**Code**. We use the code implemented by authors in github[6].

**Hyperparameters** We use the same hyperparameters summarized in Table 9 of the paper (Hou et al., 2020). We use $(m_w, m_d) = (0.25, 1.0)$ for DynaBERT-w and $(m_w, m_d) = (0.5, 0.5)$ for DynaBERT-d where $m_w$ and $m_d$ are width and depth multipliers, respectively. We do not use data augmentation for fairness since other algorithms do not use data augmentation. We report the accuracy after final-finetuning.

### C.4.4   EBERT (LIU ET AL., 2021)

**Code**. We use the code implemented by authors in github[7].

**Hyperparameters** We use the same set of hyperparameters introduced in Section 4.1 of the paper (Liu et al., 2021).

## D   RETRAINING-FREE MODEL COMPRESSION IN CNN

There are retraining-free structured pruning algorithms (YVINEC et al., 2021; Kim et al., 2020; Srinivas & Babu, 2015) for CNNs which reduce the size of pre-trained models by finding similar neurons based on their weight distribution, and integrating the similar neurons. However, we do not compare them with K-prune since they are not directly applicable to the PLM compression problem. The main reason is the architectural difference between CNN and Transformer. The structured pruning algorithms for CNN do not consider pruning of attention heads, and thus they can prune only FFN sub-layers like KCM (Nova et al., 2023) which shows severe accuracy degradation in Figure 2 of our main text.

## E   EXPERIMENTS ON LARGE LANGUAGE MODELS

We provide an experimental result on decoder-based large language models (LLMs) considering the growing interest in reducing the cost of LLMs via compression (Park et al., 2024). We prune OPT-1.3B and OPT-2.7B models (Zhang et al., 2022) using 128 sentences in C4 dataset Raffel et al.

---

[6]https://github.com/huawei-noah/Pretrained-Language-Model/tree/master/DynaBERT
[7]https://github.com/zejiangp/EBERT

(2020) and measure the perplexity on Wiki-text2 (Merity et al., 2017) dataset for evaluation. We summarize the experimental results in Table 6. Lower perplexities mean better results.

Table 6: Perpelxities on Wiki-text2 dataset (Merity et al., 2017) of OPT (Zhang et al., 2022) models pruned by K-prune. The term "Difference" represents the ratio of the amount of increased perplexity after pruning compared to the perplexity of the unpruned model, i.e. (Difference) = ((perplexity after pruning) - (perplexity before pruning))/(perplexity before pruning) $\times$ 100.

| OPT-1.3B | | | | | |
|---|---|---|---|---|---|
| Pruning rate | 0% | 5% | 10% | 15% | 20% |
| Perplexity | 14.67 | 14.41 | 13.96 | 14.67 | 15.74 |
| Difference | - | -1.77% | -4.84% | 0.00% | 7.29% |
| OPT-2.7B | | | | | |
| Pruning rate | 0% | 5% | 10% | 15% | 20% |
| Perplexity | 12.46 | 12.23 | 11.94 | 12.01 | 12.51 |
| Difference | - | -1.85% | -4.17% | -3.61% | 0.40% |

As a result, K-prune successfully prunes billion-scale LLMs maintaining its performance. Surprisingly K-prune shows negligible performance degradation of 0.4% for OPT-2.7B under 20% pruning rate. Note that structured pruning of decoder-based language models is much more difficult than that of encoder-based models. For example, LLM-pruner (Ma et al., 2023) shows severe performance degradation of over 30% for LLaMA-7B models on Wiki-text2 dataset. Combined with the observation that larger models are easier to prune (Frantar & Alistarh, 2023), we expect that K-prune achieves higher pruning rates with minimal performance degradation for language models larger than 2.7B. Therefore, applying K-prune to decoder-based LLMs is a promising future work.

