# OpenReview forum: "Accurate Retraining-free Pruning for Pretrained Encoder-based Language Models"
_ICLR.cc/2024/Conference — ICLR 2024 poster_

### Official Review · Reviewer_nct7 · 2023-10-27

**Soundness:** 4 excellent
**Presentation:** 3 good
**Contribution:** 4 excellent
**Rating:** 8
**Confidence:** 5

**Summary:**

This paper presents a method to prune an encoder-only pre-trained transformer-based language model without retraining it. The method is based on the notion of a global importance score for each attention head or feed-forward neuron, considering both the prediction loss (KL divergence between pruned and unpruned models) and the representational loss (L2 distance between weights at each layer before and after pruning). It uses a layerwise pruning of attention heads and feed-forward neurons, starting with the bottom sublayer and working up. In each layer, three steps are performed: 1) Global importance computation: Compute the global importance score for each attention head or neuron. 2) Mask search: Search for a mask for each attention head or neuron, considering both prediction and representational knowledge. 3) Linear projection layer tuning: Tune the linear projection layers for the heads or neurons that are not pruned using a linear solver on a small amount of data. The proposed approach achieves an improvement between 8.5-58.0%p in F1 across tasks/models relative to a training-free approach in Kwon et al. (2022). Compared to the unpruned baseline, the proposed approach achieves an inference speedup of 2.65x, while the method of Kwon et al. gives at most a 1.87x speedup. It achieves comparable accuracy to retraining approaches (such as DynaBERT) with significantly less computation.

**Strengths:**

*  Presents a new method for pruning a pre-trained encoder-only transformer language model without retraining.

*  The proposed approach achieves F1 score improvements of 8.5% to 58% over existing training-free pruning approaches, while achieving similar performance to retraining approaches at a much lower computational cost.

* Reports an ablation study that shows the relative importance of each component of the approach, revealing that layerwise pruning and weight tuning are critical.

**Weaknesses:**

* Approach is restricted to encoder only models but many of the current LLM approaches are decoder-based.
* Some aspects of the paper are unclear. See below in questions.

Update after author rebuttal:
* The authors have clarified most of my questions in their rebuttal.

**Questions:**

* Sec 1 : What does the "%p" notation mean?
* Sec 2.2: "Once the mask variables are established after mask search, pruning of attention heads and neurons whose mask variables are zero does not affect the inference results." What does 'established' mean in this context?
* Equation 7: The LHS K_{rep,l}(X_{\tau,l},X_{\rep,l};0)  should be a function of i.
* Sec 3.3: "We estimate the importance score not only in the target sublayer but also in the sublayers above the target sublayer" Why is it necessary to estimate the importance score for sublayers above the target sublayer considering that only the heads/neurons in the target sublayer are pruned when a given layer is considered i.e. From the description/Figure 2, it seems like Algo 1 is run for each sub-layer.
* The algorithm performs sublayer-wise pruning from the bottom to the top sub-layer. If we refer to this as a single-pass, is it ever necessary in practice to do a second pass to achieve the desired number of FLOPs?

---

> ### Author Response · Authors · 2023-11-17
>
> > **[Q1] Approach is restricted to encoder-based models but many of the current LLM approaches are decoder-based.**
>
> **[A1]** Theoretically, Kprune is applicable to the decoder-based models since decoder-based Transformers also consist of multi-head attention (MHA) and feedforward network (FFN) sub-layers; we are able to measure the amount of knowledge in each attention head and neuron.
> In this paper, we focus on encoder-based models to exhaustively demonstrate the effectiveness of Kprune since there are both state-of-the-art retraining-free and retraining-based algorithms for encoder-based models; we demonstrate the accuracy of Kprune by comparing it to retraining-free pruning algorithms and show the efficiency of Kprune by comparing it to retraining-based algorithms. On the other hand, it is impossible to compare Kprune to the retraining-based pruning algorithms using LLMs since retraining-based pruning algorithms require intractable time to prune LLMs.
> As we mentioned in the Conclusion, extending Kprune to decoder-based models is an interesting future work.
>
> > **[Q2] Sec 1: What does the "%p" notation mean?**
>
> **[A2]** “%p” is a symbol of percent-point which is used for representing the difference of percentage values. The term percent-point is distinct from percent (%) though both terms signify shifts in percentage values. Percent (%) expresses the proportional change between two numbers, while the percent-point represents the absolute difference. For instance, a 10% increase from 50% results in 55%, whereas adding 10 percentage points (%p) to 50% yields 60%. We insert a footnote in Section 1 about %p for clarity.
>
> > **[Q3] Sec 2.2:  What does 'established' mean in this context?**
>
> **[A3]**  In this context, ‘the mask variables are established’ means that the values of the mask variables are determined. We aim to deliver that imposing a mask variable as zero is equal to the pruning of the corresponding attention head or neuron since the output of MHA or FFN is represented as the masked summation of attention heads or neurons, respectively. We substitute the word “established” with “determined” for clarity.
>
> > **[Q4] Equation 7: The LHS $K_{rep,l}(X_{\mathcal{T},l},X_{\mathcal{S},l};0)$ should be a function of $i$.**
>
> **[A4]** We use the notation $K_{rep,l}(X_{\mathcal{T},l},X_{\mathcal{S},l};0)$ as the simplified notation of $K_{rep,l}(X_{\mathcal{T},l},X_{\mathcal{S},l};m_{l,i}=0)$ which is a function of $i$. We substitute the simplified notation into $K_{rep,l}(X_{\mathcal{T},l},X_{\mathcal{S},l};m_{l,i}=0)$ to clearly show that $K_{rep,l}$ is a function of $i$ in Sections 3.2 and B.1.
>
> > **[Q5] Sec 3.3: "We estimate the importance score not only in the target sublayer but also in the sublayers above the target sublayer" Why is it necessary to estimate the importance score for sublayers above the target sublayer considering that only the heads/neurons in the target sublayer are pruned when a given layer is considered i.e. From the description/Figure 2, it seems like Algo 1 is run for each sub-layer.**
>
> **[A5]** Kprune measures the importance scores of the target sublayers and upper sublayers to determine the number of attention heads and neurons to prune in each sublayer, considering their global importance. If we do not measure the importance scores of masked units (attention heads or neurons) in upper sublayers, we cannot estimate how important the target sublayer is in the entire model. Therefore, we have to prune each sublayer equally, without considering the sublayer-wise importance, resulting in a significant accuracy drop.
> In Figure 2, our target sublayer is the 2nd sublayer and we use the importance score of the 3rd and the 4th sublayers to determine the amount of neurons to prune (b). As illustrated in (c) we prune neurons only in the 2nd sublayer (target), not in the 3rd and the 4th sublayers even though we measure the importance of masked units in the 3rd and the 4th sublayers. Then, we move on to the next iteration to prune attention heads in the 3rd sublayer.
> We revise the description of Figure 2 in Section 3.1 to clarify the reason why we estimate the importance scores in the upper sublayers.
>
> > **[Q6] The algorithm performs sublayer-wise pruning from the bottom to the top sub-layer. If we refer to this as a single pass, is it ever necessary in practice to do a second pass to achieve the desired number of FLOPs?**
>
> **[A6]** You don’t need to manually tune hyperparameters or re-run Kprune to match the desired number of FLOPs. Kprune automatically finds the global pruning configurations that match the pre-defined desired pruning rate by adjusting the FLOPs budget after pruning each sublayer. We revise the last sentence in Section 3.4 to clearly deliver that Kprune does not require user intervention, such as hyperparameter tuning, to match the desired number of FLOPs.

---

> > ### Comment · Reviewer_nct7 · 2023-11-21
> > **Thanks for the explanation**
> >
> > Thanks for your explanations!

---

> > > ### Author Response · Authors · 2023-11-23
> > >
> > > We are truly delighted that our response satisfactorily addresses your concern. We sincerely appreciate your time for reviewing and providing valuable feedback.

---

### Official Review · Reviewer_sVyn · 2023-10-31

**Soundness:** 3 good
**Presentation:** 3 good
**Contribution:** 3 good
**Rating:** 6
**Confidence:** 4

**Summary:**

The paper introduces Kprune (Knowledge-preserving pruning), a novel retraining-free structured pruning algorithm designed for pretrained encoder-based language models. The key challenge addressed by this work is the accurate compression of such models without requiring retraining. Existing retraining-free algorithms suffer from significant accuracy degradation, particularly at high compression rates, due to their inability to handle pruning errors effectively.

Kprune employs an iterative pruning process that focuses on preserving the valuable knowledge contained within the pretrained model. This process includes three main steps: knowledge measurement, knowledge-preserving mask search, and knowledge-preserving weight-tuning. By implementing these steps, Kprune achieves remarkable results, with accuracy improvements of up to 58.02% higher F1 score compared to existing retraining-free pruning algorithms. These improvements are observed under a high compression rate of 80% when tested on the SQuAD benchmark, all without the need for any retraining.

**Strengths:**

1. Kprune presents a significant advancement in the field of pretrained language model compression by demonstrating the feasibility of high compression rates without compromising model accuracy. The approach's success lies in its ability to preserve the essential knowledge within the model, ultimately leading to impressive gains in performance compared to existing retraining-free pruning methods.
2. The equations are clearly described.

**Weaknesses:**

1. The writing of the introduction section seems unreasonable. I think some challenges and other content should be written in the introduction instead of the method.
2. In the main experiment table (Table 1), there are not enough baselines for comparison.

**Questions:**

Please refer to Weakness

---

> ### Author Response · Authors · 2023-11-17
>
> We sincerely appreciate the reviewer’s careful feedback. We respond to your feedback below.
>
> > **[Q1] The writing of the introduction section seems unreasonable. I think some challenges and other content should be written in the introduction instead of the method.**
>
> **[A1]** Thank you for your constructive review. We have revised the Introduction section to reflect your comment. We add a paragraph about the need for retraining-free pruning algorithms and clarify the limitations of existing retraining-free algorithms.
>
> > **[Q2] In the main experiment table (Table 1), there are not enough baselines for comparison.**
>
> **[A2]** To the best of our knowledge, Kwon et al. (NeurIPS’22) and KCM (ICML’23) are the best-performing retraining-free pruning algorithms for encoder-based models, and Kprune is the first algorithm that outperforms both. In detail, Kwon et al. is the first retraining-free pruning algorithm for PLMs and Kwon et al. is the only competitor for KCM. There are a few retraining-free algorithms for (decoder-based) LLMs, such as SparseGPT[1] or Wanda[2], but, they employ an impractical unstructured pruning pattern; they require inefficient sparse tensor representations for memorization and customized hardware for acceleration.
>
> **[Reference]**
>
> [1] Frantar, Elias, and Dan Alistarh. "SparseGPT: Massive Language Models Can Be Accurately Pruned in One-Shot." (2023).
>
> [2] Sun, Mingjie, et al. "A Simple and Effective Pruning Approach for Large Language Models." arXiv preprint arXiv:2306.11695 (2023).

---

> > ### Comment · Reviewer_sVyn · 2023-11-23
> > **Official Comment by Reviewer sVyn**
> >
> > Thanks for the detailed response. I have read the rebuttal and most of my concerns have been well addressed. Overall, I am towards acceptance and will maintain my score.
> >
> > Best, The reviewer sVyn

---

### Official Review · Reviewer_9HrX · 2023-10-31

**Soundness:** 3 good
**Presentation:** 3 good
**Contribution:** 3 good
**Rating:** 8
**Confidence:** 4

**Summary:**

The paper presents KPrune, a new retraining-free structured pruning method for compressing task-specific models while retaining their knowledge. Unlike prior techniques, KPrune considers both the overall loss impact and the effect on representation preservation when deciding which units to prune. It does this by analyzing the expected output of each head/neuron. As a result, KPrune requires minimal pruning time compared to retraining methods and outperforms previous retraining-free approaches. This work differs from Kwon et al. (2022) in two main ways: 1) it incorporates layer-wise representation loss when measuring unit importance, and 2) it recovers weights by tuning the output matrix with linear solvers. These innovations enhance pruning quality with minimal overhead versus retraining. In summary, KPrune efficiently compresses models while maintaining performance by carefully accounting for loss impact and representation preservation when pruning.

**Strengths:**

- The paper is well written and easy to follow!
- KPrune is the first retraining-free pruning method to incorporate layer-wise representation preservation loss and KL loss on outputs, techniques commonly used in retraining-based pruning.
- KPrune shows considerable performance improvements over previous retraining-free methods in high sparsity regimes (>80%). This demonstrates its ability to maintain model quality even with extreme compression rates.

**Weaknesses:**

- While not a direct comparison, it would be interesting to compare KPrune's performance to stronger training-based methods like CoFiPruning. KPrune's  major advantage is requiring much less time for pruning, though training-based approaches may achieve better end results.
- Since BERT models are relatively small in scale nowadays, reducing training compute is less impactful compared to large language models. It would be interesting to evaluate if KPrune can effectively scale up to larger and more powerful models where saving compute will be more significant.

**Questions:**

I don't have further questions for the paper.

---

> ### Author Response · Authors · 2023-11-17
>
> We sincerely appreciate the reviewer’s careful feedback. We respond to your feedback below.
>
> > **[Q1] While not a direct comparison, it would be interesting to compare KPrune's performance to stronger training-based methods like CoFiPruning[1]. KPrune's major advantage is requiring much less time for pruning, though training-based approaches may achieve better end results.**
>
> **[A1]** Thank you for suggesting an interesting experimental setup. However, as we mention in the “Competitors” paragraph in Section 4.1 of our paper, there is a reason for excluding CoFi as a competitor. According to Kwon et al.[2],  CoFi shows similar or worse accuracy than DynaBERT (Figure 6 in Kwon et al.) while requiring more than 2.7 times longer time for pruning than DynaBERT (Table 2 in Kwon et al.). Therefore, we chose DynaBERT and EBERT as our competitors and conducted our experiments accordingly.
>
> > **[Q2] Since BERT models are relatively small in scale nowadays, reducing training compute is less impactful compared to large language models. It would be interesting to evaluate if KPrune can effectively scale up to larger and more powerful models where saving compute will be more significant.**
>
> As noted in the response [A2] for the reviewer **[akzo]**, the cost of Kprune for a model with $L$ sublayers is equivalent to performing $(L+1)/2$ times of forward and backward propagations on tiny sample data. If we follow the setting of SparseGPT[1], an “unstructured” pruning algorithm for large language models (LLMs), this involves using only about 128 sample instances. Therefore, if the hardware setting is adequate for running inference on massive LLMs, it is entirely feasible to execute Kprune within a tractable amount of time. Extending accurate and efficient Kprune to massive LLMs is our promising future work.
>
> **[Reference]**
>
> [1] Xia, Mengzhou, Zexuan Zhong, and Danqi Chen. "Structured pruning learns compact and accurate models." Annual Meeting of the Association for Computational Linguistics (2022)”: 1513-1528
>
> [2] Kwon, Woosuk, et al. "A fast post-training pruning framework for transformers." Advances in Neural Information Processing Systems 35 (2022): 24101-24116.

---

> > ### Comment · Reviewer_9HrX · 2023-11-20
> > **Inference latency of the pruned models**
> >
> > Thanks for your response! I noticed that this paper, as well as Kwon et al., 2022, both use reduced FLOPs instead of inference latency as the metric to compare against different approach and it might be misleading. Reduced FLOPs and inference runtime could lead to very different results and interpretations of the final outcome of a pruning approach. Adding wall-clock analyses would be very helpful for understanding the effectiveness of the approach.

---

> > > ### Author Response · Authors · 2023-11-23
> > >
> > > We sincerely appreciate the valuable feedback regarding the computational cost of Kprune. As you mentioned, wall clock time analysis is crucial in model compression, and we conduct an experiment that compares the inference speed (wall clock time for inference) of models compressed by Kprune (ours) and competitors. The experimental results are summarized in Table 1 of the main text. The experimental results show that models pruned by Kprune are accelerated by up to 2.93 times, which is significantly higher than the maximum acceleration of 2.1 times achieved by competitors. Therefore, we demonstrate that reducing FLOPs using Kprune leads to inference speedup of the pruned model.

---

### Official Review · Reviewer_akzo · 2023-11-02

**Soundness:** 2 fair
**Presentation:** 2 fair
**Contribution:** 2 fair
**Rating:** 5
**Confidence:** 4

**Summary:**

This paper introduces a new algorithm called Kprune that can significantly improve the accuracy of pretrained language models while compressing them without the need for retraining. The authors explain that while pruning is a common technique for compressing deep neural networks, it is often difficult to apply to pretrained language models due to their complex architecture and the difficulty of preserving their useful knowledge during the pruning process. Kprune addresses these challenges by using an iterative pruning process that selectively removes neurons from the model based on their importance to the overall performance of the model. The authors evaluate Kprune on several benchmark datasets and compare its performance to existing retraining-free pruning algorithms.

**Strengths:**

- The paper introduces a new algorithm, Kprune, that can significantly improve the accuracy of pretrained language models while compressing them without the need for retraining.
- The authors provide a detailed explanation of the iterative pruning process used in Kprune and how it helps preserve the useful knowledge of the pretrained model.
- The authors evaluate Kprune on several benchmark datasets and compare its performance to existing retraining-free pruning algorithms, providing evidence of its effectiveness.
- The paper could have practical applications in the field of natural language processing, where the ability to compress pretrained language models without sacrificing accuracy is highly desirable.

**Weaknesses:**

- The paper focuses specifically on encoder-based language models, so it may not be applicable to other types of language models.
- The authors do not provide a detailed analysis of the computational resources required to implement Kprune, which could be a potential limitation for some applications.
- The paper does not apply to large language models which are commonly used now, experiments on larger models are expected.

**Questions:**

Can the proposed technique applied to LLMs?

---

> ### Author Response · Authors · 2023-11-17
>
> We sincerely appreciate the reviewer for expressing your concerns. We respond to your concerns and questions below.
>
> > **[Q1] The paper focuses specifically on encoder-based language models, so it may not be applicable to other types of language models.**
>
> **[A1]** Theoretically, Kprune is applicable to the decoder-based models since decoder-based Transformers also consist of multi-head attention and feedforward network sub-layers; we are able to measure the amount of knowledge in each attention head and neuron.
> On the other hand, we focus on encoder-based models to exhaustively demonstrate the effectiveness of Kprune since there are both state-of-the-art retraining-free and retraining-based algorithms for encoder-based models; we demonstrate the accuracy of Kprune by comparing it to retraining-free pruning algorithms and show the efficiency of Kprune by comparing it to retraining-based algorithms. On the other hand, it is impossible to compare Kprune to the retraining-based pruning algorithms using LLMs since retraining-based pruning algorithms require intractable time to prune LLMs. As we mentioned in the Conclusion, extending Kprune to decoder-based models is an interesting future work.
>
> > **[Q2] The authors do not provide a detailed analysis of the computational resources required to implement Kprune, which could be a potential limitation for some applications.**
>
> **[A2]** The computational cost of Kprune is composed of the computational costs of (a) Knowledge measurement, (b) Knowledge-preserving mask search (KPMS), and (c) Knowledge-preserving weight-tuning processes (KPWT). During (b) KPMS, the process involves inexpensive calculation of importance scores based on the measured amount of knowledge, and comparison of the scores to find the pruning masks; thus KPMS  takes a very short time. Moreover, (c) KPWT is also an inexpensive process, completing within a second for all sublayers when using the “lstsq” solver provided in PyTorch. Therefore, the computational cost of Kprune mainly depends on the (a) knowledge measurement process, which involves repetitive forward and backward operations of sublayers.
>
> If we assume that $M$ is the number of operations required for forward and backward propagations in a model with $L$ sublayers, then the operations required per sublayer are considered as $M/L$. The knowledge measurement process includes forward and backward operations for each sublayer from the bottom to the top. Each operation includes $L$, $L-1$, ..., and $1$ times of forward and backward operations of sublayers, and requires $(L+1)/2 * M$ operations in total, i.e. it is the same as the cost of running $(L+1)/2$ epochs on a sample dataset. Combined with the fact that Kprune utilizes a tiny sample dataset (about 0.64% of the MNLI dataset), the cost of Kprune is significantly low and Kprune is scalable to the larger language models.
> In our paper, Kprune takes a few minutes on a single NVIDIA 1080Ti GPU for pruning BERT-base which is up to 422 times shorter time for pruning retraining-based algorithms.
>
> > **[Q3] The paper does not apply to large language models which are commonly used now, experiments on larger models are expected.
> Can the proposed technique applied to LLMs?**
>
> **[A3]** As noted in [A2], the cost of Kprune for a model with $L$ sublayers is equivalent to performing $(L+1)/2$ times of forward and backward propagations on a tiny sample dataset. If we follow the setting of SparseGPT[1], an “unstructured” pruning algorithm for large language models (LLMs), this involves using only about 128 sample instances. Therefore, if the hardware setting is adequate for running inference on massive LLMs, it is entirely feasible to execute Kprune within a tractable amount of time. Extending accurate and efficient Kprune to massive LLMs is our promising future work.
>
> **[Reference]**
>
> [1] Frantar, Elias, and Dan Alistarh. "SparseGPT: Massive Language Models Can Be Accurately Pruned in One-Shot." (2023).

---

### Author Response · Authors · 2023-11-23
**Regarding decoder-based models**

First of all, we appreciate all reviewers for providing insightful and constructive feedback. The feedback is currently reflected in the manuscript in blue text, and the feedback significantly enhanced the quality of our work. Many reviewers have positively commented on the originality, practicality, and readability of our paper. On the other hand, some reviewers expressed curiosity about whether Kprune could be extended to decoder-based models, particularly to large language models (LLMs). Extending Kprune to decoder-based models is one of the most interesting and promising future works for us, and we perform preliminary experiments to demonstrate the potential of Kprune to be applied to decoder-based models.

We apply Kprune to OPT-125m [1] which is a widely used decoder-based model. We use 128 sentences from the C4 dataset [2] as a calibration set and measure the perplexity of the Wiki-text2 [3] dataset for evaluation [4]. We prune 5% to 20% of the parameters of the unpruned model following the setting of LLM-pruner (NeurIPS’23) [5]. We summarize the experimental results in Table 1.

**[Table 1]** Perplexity on the Wiki-text2 dataset of OPT-125m pruned by Kprune (ours). Lower perplexities represent better performances. The Diff. row represents the percentage of the amount of the increased perplexity w.r.t. the unpruned model (0%).
| Pruning rate |   0%  |   5%  |  10%  |  15%  |  20%  |
|:------------:|:-----:|:-----:|:-----:|:-----:|:-----:|
| Perplexity ↓ | 27.66 | 26.88 | 27.31 | 28.83 | 31.72 |
|   Diff. (%)  |   -   | -2.82 | -1.27 |  4.23 | 14.68 |


As a result, Kprune successfully prunes OPT-125m while increasing the unpruned model’s perplexity lower than 15% under all pruning rates. Note that retraining-free structured pruning of decoder-based models is much more difficult than that of encoder-based models; for instance, LLM-pruner (NeurIPS’23) [5], which is the state-of-the-art structured pruning algorithm for decoder-based models, shows severe performance degradation more than 30% under the pruning rate of 20% even though it performs fine-tuning after pruning. Therefore, Kprune has a potential to be applied to decoder-based LLMs, and performing extensive experiments on such settings with advanced hardwares is one of our future works.

**[Reference]**

[1] Zhang, Susan, et al. "Opt: Open pre-trained transformer language models, (2022)." URL https://arxiv.org/abs/2205.01068.

[2] Raffel, Colin, et al. "Exploring the limits of transfer learning with a unified text-to-text transformer." The Journal of Machine Learning Research 21.1 (2020)

[3] Merity, Stephen, et al. "Pointer sentinel mixture models." arXiv preprint arXiv:1609.07843 (2016).

[4] Frantar, Elias, and Dan Alistarh. "SparseGPT: Massive Language Models Can Be Accurately Pruned in One-Shot." (2023).

[5] Ma, Xinyin, et al. "LLM-Pruner: On the Structural Pruning of Large Language Models." Advances in Neural Information Processing Systems 35 (2023)

---

### Meta-Review · Area_Chair_NepK · 2023-12-06

**Metareview:**

Overall, reviewers recognize this work's potential to advance efficient deployment of language models by enabling aggressive pruning without costly retraining. The proposed iterative method outperforms prior retraining-free approaches by 8-58% in accuracy via carefully preserving model knowledge. Speedups of 2.6x are demonstrated versus unmodified models. These gains persist even under 80% sparsity levels where existing techniques degrade badly.

In light of the largely positive appraisal across criteria and authors beginning to address key weaknesses during rebuttal, I recommend acceptance to further develop a method pushing the boundaries of efficient LLM deployment. Authors are encouraged to provide continued experiments on decoder-style autoregressive transformers and bigger scales.

**Justification For Why Not Higher Score:**

Some concerns centered around decoder-only transformer applicability and scaling up to the very largest models. The rebuttal mitigates these by showing strong initial results pruning a 1.3B parameter decoder model - but scaling these experiments further is strongly encouraged. Reviewers also ask for more analysis of design choices and resource overheads.

**Justification For Why Not Lower Score:**

An important topic to work on. Multiple reviewers praised the simplicity, effectiveness, and practical appeal given the growing language model sizes. Comparisons to related training-based methods demonstrate competitive gains at massively reduced resource costs.

---

### Decision · Program_Chairs · 2024-01-16

Accept (poster)